# Research on Image Stitching Algorithm Based on Point-Line Consistency and Local Edge Feature Constraints

Shaokang Ma, Xiuhong Li *, Kangwei Liu [ID], Tianchi Qiu and Yulong Liu [ID]

Key Laboratory of Signal Detection and Processing, Department of Computer Science and Technology, Xinjiang University, Urumqi 830017, China; 107552103639@stu.xju.edu.cn (S.M.); lkw21@stu.xju.edu.cn (K.L.); qiutc@stu.xju.edu.cn (T.Q.); 107552103588@stu.xju.edu.cn (Y.L.)
* Correspondence: xjulxh@xju.edu.cn

**Abstract:** Image stitching aims to synthesize a wider and more informative whole image, which has been widely used in various fields. This study focuses on improving the accuracy of image mosaic and proposes an image mosaic method based on local edge contour matching constraints. Because the accuracy and quantity of feature matching have a direct influence on the stitching result, it often leads to wrong image warpage model estimation when feature points are difficult to detect and match errors are easy to occur. To address this issue, the geometric invariance is used to expand the number of feature matching points, thus enriching the matching information. Based on Canny edge detection, significant local edge contour features are constructed through operations such as structure separation and edge contour merging to improve the image registration effect. The method also introduces the spatial variation warping method to ensure the local alignment of the overlapping area, maintains the line structure in the image without bending by the constraints of short and long lines, and eliminates the distortion of the non-overlapping area by the global line-guided warping method. The method proposed in this paper is compared with other research through experimental comparisons on multiple datasets, and excellent stitching results are obtained.

**Keywords:** image stitching; local edge contour features; image registration; mesh optimization

## 1. Introduction

Image stitching is performed to combine images or parts of images with overlapping areas to create an image with a wider overall field of view, higher resolution and richer information. The technology is continuously updated and iterated, widely used in fields such as panoramic photography, autonomous driving, medical imaging, and virtual reality [1–3]. Pictures taken in different scenes and at different times bring large parallax, which leads to serious artifacts and perspective distortion in the stitching results. Therefore, improving the level of image processing and obtaining high-quality images has been one of the focuses of research.

Image registration is a crucial step in the image stitching process. It establishes a spatial mapping relationship between image pixels to ensure accurate alignment of different images [4]. The classical feature extraction technique [5–7] in image registration based on sparse features has been applied so far. Later, the concept of line features was introduced [8,9]. Many studies [10,11] began to explore the full integration of point and line features in image registration and transformation to achieve better alignment and more natural stitching. However, most stitching methods are based on the separate registration of images using point and line features, which cannot fully consider the overall alignment of both points and lines, leading to inaccurate matching. Jia et al. proposed a local region block strategy, which uses feature numbers to enrich matching points to improve alignment ability [12]. These methods all consider the combination of points and lines to improve alignment accuracy, but ignore special scenarios where point and line feature extraction

is limited and feature matching accuracy is low. Information entropy can describe the diversity and information content of features. Higher information entropy indicates that the image area contains more different features, while lower information entropy may indicate that a certain area has more consistent features. By analyzing information entropy distribution, we can determine which areas have rich features, and we can pay more attention to these areas during splicing.

Traditional image stitching methods [13,14] use global homography stitching techniques and often struggle to handle the alignment of local details effectively. Zaragoza et al. introduced the APAP algorithm, which applies the moving Direct Linear Transformation (M-DLT) framework for image stitching [15]. This approach divides the images into a dense grid and computes local homography transformations separately on each grid. Then, the image mosaic problem is transformed into a mesh optimization problem, and the image is aligned and adjusted more accurately by optimizing the network model. Additionally, in research, there are image stitching methods based on seam lines. Because dislocation artifacts are difficult to avoid in the overlapping area, in [16–19], methods were proposed to find the best suture in the overlapping area to deal with parallax and eliminate dislocation artifacts. However, although these methods can improve the misalignment problem, the stitched images still retain their respective viewing angles, resulting in a certain degree of single viewing angle effect in the stitched results.

Grid optimization solves the alignment problem of overlapping area well, but there are projection distortion and perspective distortion in the multi-view, which is easy to produce serious distortion in non-overlapping area. Many research efforts often employ methods that combine improved alignment in overlapping regions with constraints to minimize distortion in non-overlapping areas to address the distortion issue. The shape-preserving method introduces shape correction and image scaling to reduce deformation distortion [20]. Chen introduced global similarity, local similarity, and parallax rotation constraints to optimize the mesh [21]. Liao et al. proposed a single-view image mosaic method to prevent the linear structure from being deformed due to warpage in the image. The linear structure protection item and distortion control item designed in the energy function achieved a good stitching effect [22]. Jia et al. designed long straight lines to ensure the stability of the straight structure in the whole image [12].

In order to improve the quality and performance of stitching, many researchers have introduced the concepts of entropy and information theory into image stitching [23,24]. Among existing research, some methods utilize entropy or information theory to select appropriate features for matching. These methods usually calculate the information entropy or cross-entropy of an image to evaluate the complexity and information content of the image. In image splicing, if the entropy difference between two images is large, the difficulty of splicing them increases accordingly. Therefore, by calculating the entropy of an image, one can better understand the difficulty of stitching and the amount of information required. Information theory helps to evaluate the difference between the stitched result and the original image, i.e., using cross-entropy to measure the difference between the stitched result and the original image. The smaller the cross-entropy, the closer the image information of the splicing result is to the original image, and the better the performance of the algorithm. By comparing the cross-entropy of different splicing algorithms, their performance can be evaluated and the optimal algorithm can be selected.

In recent years, image mosaic algorithms have made remarkable progress in dealing with parallax and distortion, but they are still not comprehensive enough. In fact, there are many different types of feature structures in the image, but most of the stitching algorithms usually focus on the point and line structures and ignore the local edge contour, which can effectively reflect the whole scene structure. In this paper, a mosaic method is designed, which makes full use of the local edge structure to improve the quality of image mosaic.

Based on the research of common natural image data, this paper focuses on the image stitching problem caused by single-image warping and proposes an image mosaic algorithm based on point-line consistency and edge contour feature constraint. The final

image stitching is achieved through three steps: feature matching correspondence, image warpage and image fusion. This paper mainly studies the following three aspects:

- Aiming at the problem of insufficient features of low texture region in the overlapping region, the point-line consistency module is proposed, which uses SIFT with good stability to extract features, to increase the number of matching point pairs and filter out erroneous matches.
- Aiming at the problem of many structural features in the image that are not fully utilized, and the point-line feature being wrong, the method in this paper innovatively breaks through the traditional understanding of the structural features of the image. It not only takes into account the limitations of point and line features, but it also fully exploits the rich structural information of the image. Local edge contour features are constructed to constrain global image pre-registration, weaken the impact of mismatching, and thereby improve the accuracy of image alignment.
- Aiming at the problem of alignment and distortion imbalance in single image warpage stitching, this paper introduces multiple optimization modules to ensure image alignment and minimize the distortion of non-overlapping regions.

The remainder of the paper is structured as follows: Section 2 reviews related research in the field of image stitching, including various registration methods as well as mesh optimization methods. In Section 3, the image stitching algorithm proposed in this paper is introduced in detail, and the construction principle of edge contours and the contour feature matching mechanism are explained in depth. Section 4 describes the dataset and evaluation metrics used in the experiments, and presents the experimental results in detail. Finally, Section 5 reviews the entire study, highlights the innovativeness of our proposed edge contour constraint method, discusses the limitations of the algorithm, and proposes directions for future research.

## 2. Related Work

This paper presents an image stitching method based on point-line consistency and local edge contour feature constraints. Therefore, this section reviews the work related to feature matching and image warping in the stitching method.

In the early days, when corner detection was limited, Lowe et al. introduced the Scale-Invariant Feature Transform (SIFT) algorithm, which possesses scale, rotation and brightness invariance, and is used today [5]. In addition, there is an improved SIFT-based accelerated robustness feature algorithm (SURF), which has a faster speed and is more suitable for real-time applications [6]. The ORB algorithm combines the advantages of FAST and BRIEF to detect denser matching points in the flat texture region [7]. Later, the concept of line features was introduced, and the line structure in an image is detected and matched by line detection algorithm LSD [8] and EDLines [9], which can be used as a supplement of point feature and can enhance the reliability of image registration. Recently, Jia et al. adopted the block strategy that utilizes geometric invariants for regional matching, leading to an abundance of matching point pairs [12]. In addition, feature mismatch filtering needs to be performed to improve matching accuracy. The brute force matching algorithm is usually used for preliminary matching, but the calculation efficiency is too low and is applicable to small-scale and simple scenarios. The RANSAC algorithm [25] performs well in datasets with some noise and outliers, but it has a high computational complexity. The Multi-GS algorithm [26], combining the RANSAC idea, is suitable for multiple model fitting interior points, has higher robustness, and is suitable for complex scenarios and situations where multiple models are required. Currently, most research is based on a point-line separation approach for image registration, which results in the inability to simultaneously consider both point and line structures. This leads to the exclusion of many locally effective matches. By utilizing geometric invariants [27] to establish a relationship between point features and line features, an increase in the number of matching point pairs is achieved. This is then combined with the Multi-GS algorithm to enhance the quality of matches. Compared with the traditional methods, i.e., RANSAC and point-line separation

registration, the proposed strategy is more robust in the face of limited feature detection and wrong feature matching.

Traditional image stitching methods use global homography to warp images, but they are only suitable for ideal scenarios where scene depth remains nearly constant in the overlapping areas. The Dual Homography Warping (DHW) algorithm [13] divides the scene into background and foreground planes and aligns them separately using two homography matrices. The Sliding Variational Affine (SVA) algorithm [14] improves local transformations and alignment through multiple affine transformations. In recent years, many studies have used multiple homogeneous spatial variation warping methods to deal with parallax problems. The APAP algorithm [15] divides the image into unified mesh units and obtains local homography transformation to guide image stitching, which can better deal with the parallax problem. The Adaptive As Natural As Possible warping algorithm (AANAP) [28] combines local homography transformations with global similarity transformations to effectively reduce distortion in non-overlapping regions. The Shape-Preserving Half-Projection Warping algorithm (SPHP) [20] employs subregion warping to smooth the transition between overlapping and non-overlapping regions, reducing distortion. At the same time, the image stitching method based on seam line solves the parallax problem by finding the best seam line in the overlapping area. Gao et al. proposed a seam drive method to deal with parallax and eliminate dislocation artifacts [16]. Parallax-tolerant image stitching [17,29] improves stitching performance in large parallax scenes using content preservation warpage and optimization of seam lines. Joint Guided Local Alignment (SEAGULL) [18] adds structural protection constraints for bends and straightness. Perception-based seam tailoring [19] introduces human perception and saliency detection to obtain the best seams, making the stitching results look more natural.

In image warpage processing, it is important to preserve the original state of the prominent structures in the image. The Double Feature Warp Algorithm (DFW) [10] introduces line features and constrains both points and lines to ensure that the integrity of important structures is maintained after the image is warped. The Global Similarity Prior Algorithm (GSP) [21] makes full use of line features to restrict the angle of global similarity, combining local and global constraints to reduce structural distortion. The Quasi-Homography Warping (QHW) algorithm [30] maintains the properties of quasi-homography warping by introducing intersecting lines, further enhancing the alignment of images. The Single Viewpoint Warping (SPW) algorithm [22] introduces multiple protection terms and uses the least squares method to optimize the objective function, obtaining the best homography warping transformation and thus reducing distortion. The shape structure protected stitching method (LCP) [12] introduces the concept of a global line feature, i.e., short line segments are combined into long line segments to reduce the bending of the line structure in image stitching, thereby improving the stitching effect. Together, these methods aim to maintain the integrity and accuracy of image structures in image warpage processing.

Image stitching methods based on deep learning have also produced extensive research results. The viewless image splicing network [31] proposes a view-free image splicing network. In order to reduce artifacts as much as possible, a global correlation layer and a gradual splicing module from structure to content are designed. However, the dataset it synthesizes and uses does not have parallax. Ref. [32] proposed an unsupervised deep image stitching network that is adapted to large baseline scenes and eliminates artifacts caused by pixel levels by reconstructing features. But as the disparity increases, the burden of reconstructing the network also becomes heavier. PWW [33] proposes an image stitching network composed of a pixel-by-pixel warping module (PWM) and a stitched image generation module (SIGMo) to deal with the large parallax problem by estimating the distortion in the pixel direction. The pixel-by-pixel image stitching network [34] adopts a large-scale feature extractor and an attention guidance module to achieve high-resolution and accurate pixel-level offset. A series of constraints are introduced to enhance the consistency of features. Content and structure between image pairs and stitched images.NIS [35] achieves

high-frequency details without point-of-view image stitching and blends color mismatches and misalignments to relax parallax errors.

The learning-based splicing method realizes automatic feature learning, end-to-end training and global information synthesis through deep learning networks, thereby improving the robustness and generalization ability of image splicing, especially in handling complex scenes. However, traditional methods still have their advantages in scenarios with limited resources and high real-time performance requirements. Future research may explore combining the advantages of both to achieve a more comprehensive and efficient image stitching technology.

## 3. Materials and Methods

This chapter designs and introduces the image stitching algorithm based on local edge contour feature constraints. The algorithm's stitching process is illustrated in Figure 1. The algorithm mainly includes the following stages: first, using geometric invariants [27] (i.e., feature number construction), matching point pairs are added and refined. Next, edge contours in the stitched images are detected using edge detection methods [36,37], and subsequently, local edge contour features are extracted and matched from the obtained detection results. Then, local edge contour features are used to constraint global allocation and perform grid optimization. Finally, a linear fusion method is used to fuse the image.

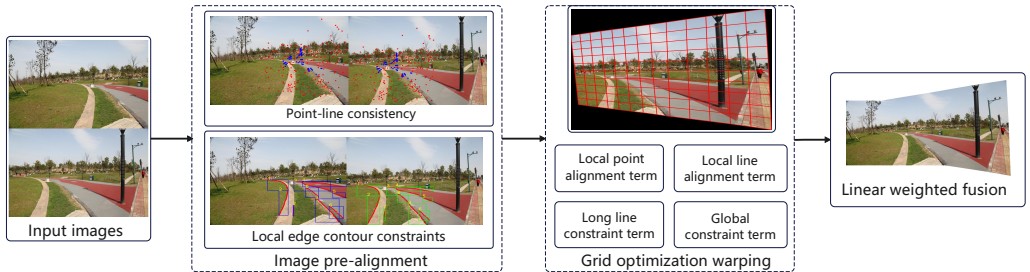

**Figure 1.** Workflow of the image stitching algorithm introduced in this paper. The stitching process is divided into three stages, including image pre-alignment, grid optimization warping and image fusion.

Figure 2 clearly shows the various steps and the operation sequence of the image registration and image optimization warping modules, where a is the flow chart of the feature extraction module and b is the flow chart of the mesh optimization warping module. At the same time, the following content provides a detailed introduction around these two main modules.

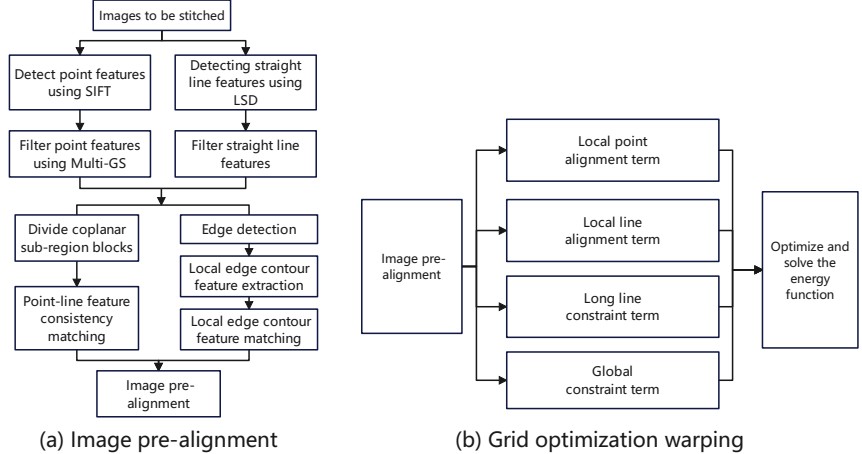

**Figure 2.** Workflow of the image stitching algorithm introduced in this paper. Flowchart (**a**) and flowchart (**b**) are detailed expansions of the image pre-alignment stage and grid optimization warping stage respectively.

### 3.1. Feature Detection and Point-Line Consistency Matching

The algorithm in this paper adopts the point-line consistency method [12] in the feature detection stage. In low-texture areas, the point features extracted by traditional point feature extraction algorithms are very limited, and the feature extraction process is prone to noise and unstable features. In the straight line detection stage, the LSD straight line detection algorithm is selected. By setting a threshold, straight lines of insufficient length are eliminated, and then consistent matching of point and line features is performed. The obtained feature point pairs are matched by adjacency relationship and then filtered using the Mutil-GS algorithm for interior points. As shown in the point-line feature consistency matching results in Figure 3, the number of feature points after point-line feature consistency expansion increases a lot, thereby better improving the accuracy of subsequent registration and alignment.

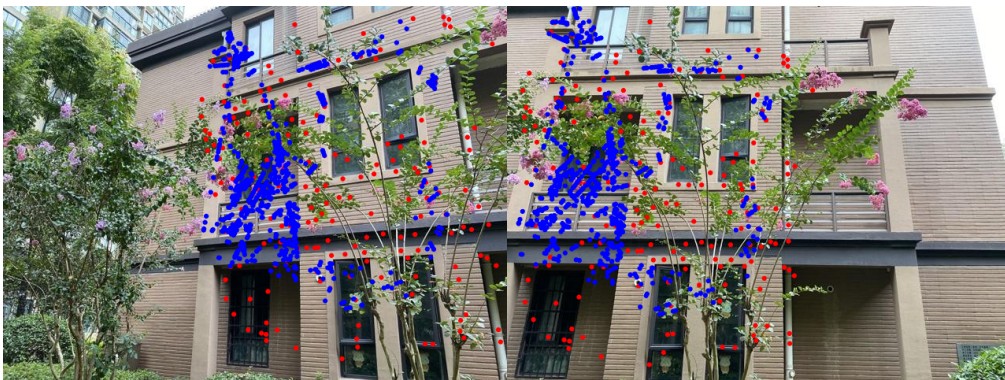

**Figure 3.** Point-line feature consistency matching results. The red points are feature points extracted and filtered by the SIFT algorithm, and the blue points are matching feature points that were increased by point-line consistency.

### 3.2. Local Edge Contour Feature Extraction

This section proposes a local edge contour feature extraction algorithm to extract local edge contour features in the image to constrain the global alignment of the image. The algorithm is mainly divided into three key steps, including edge detection, structure separation and local edge contour merging. The edge detection in this section uses the relatively stable canny algorithm [36] to generate edge detection results composed of multiple continuous local edge objects on the image to be spliced. As shown in Figure 4a, the next two key steps are performed on the results of canny detection.

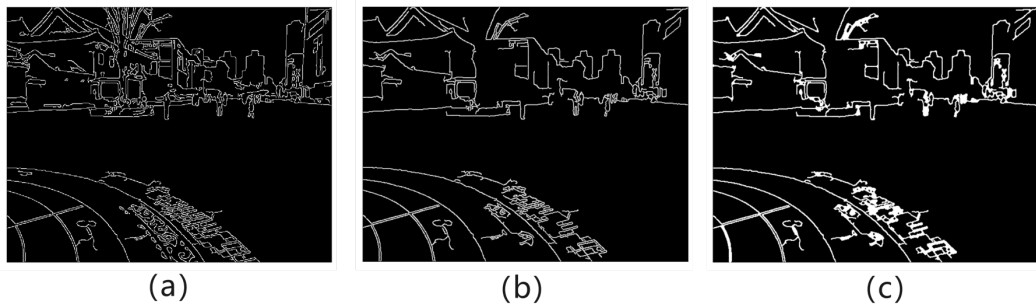

(a)        (b)        (c)

**Figure 4.** (**a**) Canny-based edge detection results. (**b**) Filtered significant and continuous local edge contour results. (**c**) Closing operation results.

#### 3.2.1. Structure Separation

The main goal of this process is to separate local edge objects that conform to quadratic curves from edge detection results. The isolated local edge objects conform to straight-line structures or approximate parabolic curve structures, so that the structure can be further analyzed and processed.

First, significant and continuous local edge objects are screened out based on the local area, pixel intensity distribution and other attribute conditions of the local edge objects, as shown in Figure 4b. Then, a closed operation is used to process each local edge object, as shown in Figure 4c. The two basic morphological operations of dilation and erosion are used to fill holes in edges, connect disconnected edges, and smooth irregular edges. The closing operation greatly improves the continuity of the edge, reduces noise, makes the edge more stable, and provides support for the implementation of subsequent contour tracking methods.

During the edge separation process, the point set and minimum bounding rectangle of each local edge object need to be recalculated after each operation.

For each local edge object, the "crawler" method is used in contour tracking to perform separation operations in the X and Y axes. A point storage matrix needs to be created for the isolated new local edge contour. The starting point is freely selected along the left boundary of the bounding box and is placed into the point storage matrix for initialization. For each unit length along the X-axis direction, the Euclidean distance between the y-value point and the last point in the point storage matrix is calculated to measure its proximity to the last point in the current point storage matrix. A point distance threshold is established, and the calculated Euclidean distances are compared with the point distance threshold. If the distance is less than the threshold, the point is included in the point storage matrix; otherwise, it continues to move forward to explore the next unit point. If a suitable point cannot be found for three consecutive times, the point set in the point storage matrix is considered to be a new local edge object. At the same time, the point set of the separated local edge object is removed from the point set of the old local edge object. Subsequently, the minimum bounding rectangle of the newly recalculated local edge objects is determined. This process is iterated until the point set of the old local edge object is empty. The above is the operation of separating local edge objects in the X-axis direction, and the principle is the same in the Y-axis direction. Finally, a collection of local edge objects in the X and Y directions is obtained. Figure 5 illustrates the process of separating a local edge object (red box) in the X direction to generate multiple local edge objects.

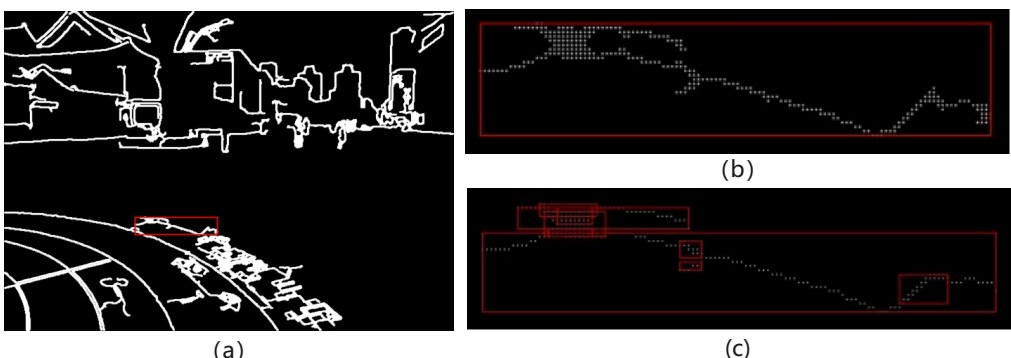

**Figure 5.** Results of separation of local edge objects along the X direction. (**a**) The red box is a local edge contour. (**b**) The red local contour shape is enlarged. (**c**) The result of separation in the X-axis direction.

The obtained local edge objects undergo direction-specific local peak detection along the coaxial directions during separation along the X and Y directions. This is because the outline structure of most local edge objects does not conform to the quadratic curve, such as the entire curve structure in Figure 6. It is necessary to detect the contour structure conforming to the quadratic curve from the curved structure and peel it off. Obviously, the more complex curve structure contains more peak points. The method of local extremum detection is employed to precisely identify peaks within the curve structure. Figure 6 is the result of peak detection on the curve structure of the largest edge separated in the X direction in Figure 5c. The curve structure of this maximum edge is represented as $P_e = ((x_1, y_1), (x_2, y_2), \ldots, (x_n, y_n))$. Then, $P_e^j = (y_1, y_2, \ldots, y_n)$ represents

the y value of the point set in the curve structure, and the peak point is calculated through the following formula :

$$P_e^{j-1} < P_e^j \quad \& \quad P_e^j > P_e^{j+1},$$
$$P_e^{j-1} > P_e^j \quad \& \quad P_e^j < P_e^{j+1}. \tag{1}$$

In the formula, the qualified data points are local high peak value and local low peak value, respectively. Finally, the decomposed smaller structures are filtered by setting a spindle length threshold.

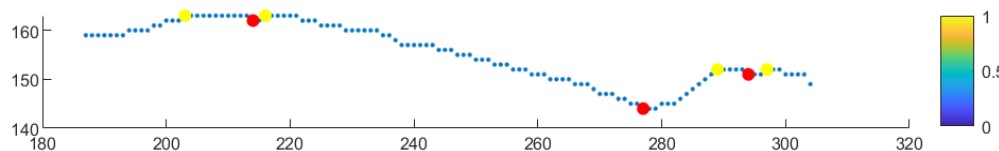

**Figure 6.** Results of peak detection on the edge contours, with yellow denoting peaks and red denoting valleys.

After each local edge object is separated, it produces overlapping or similar structures, as shown in Figure 7a where Edges 1 and 4 overlap with Edge 5. This situation is mainly caused by two factors. First, the closing operation causes the edges to be filled in and become more continuous. At the same time, the point data are also more dense, and repeated structures appear when crawling new local edge objects. Second, the new local edge objects separated in the X and Y directions have similar structures. In order to deal with this problem, the similarity or overlap degree of two edges is determined by calculating the minimum distance between two edges, and a edge similarity threshold is set to deal with dense or overlapping edge structures. We assume contour $A = \{(x_i, y_i)\}_{i=1,2,...,n}$ and contour $B = \{(u_i, v_i)\}_{i=1,2,...,m}$. For each point $(x, y) \in A$, we calculate its Euclidean distance from all points $(u, v)$ in B, and the minimum value is the distance from the point to Edge B. The distance from the point to the edge is given by the following formula:

$$D(a) = \min \left\{ \sqrt{(x - u)^2 + (y - v)^2} \quad for \quad all \quad (u, v) \in B \right\}. \tag{2}$$

Next, the distances from all points $a \in A$ to Edge B are calculated, and their mean value is computed. This mean distance serves as the minimum distance between the two edges, denoted as $D(A, B)$. The calculation formula is as follows:

$$D(A, B) = \frac{1}{n} \sum D(a) \quad for \quad all \quad a \in A. \tag{3}$$

When the distance value between two edges is smaller, it indicates that the similarity or overlap between the two edges is higher, and vice versa. Figure 7b is the result after removing overlapping or similar structures.

However, for local edge objects with intersection and non-intersection parts, directly calculating the distance between two edges cannot accurately describe the relationship between them. Here, the semi-merged separation method is used to make judgments on the intersection data. First, according to the minimum circumscribed rectangle of the local edge object, the intersection area data of the two local edge objects are obtained, and the edge distance of the intersection part is calculated. If the calculation result is less than the edge similarity threshold, the non-intersection data of the smaller edges are merged into the larger edge data; otherwise, they are regarded as local edge objects that do not interfere with each other.

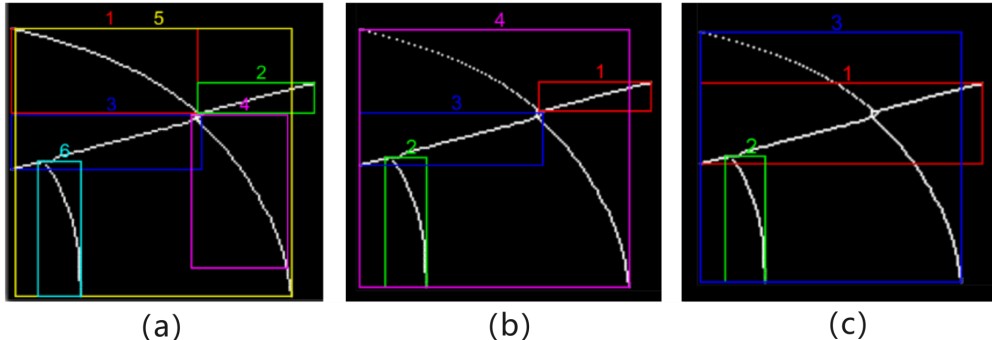

**Figure 7.** The process of structure separation and approximate merging for axis-aligned contours (local) is illustrated, with each edge numbered separately. The edge in the rectangular box represents a local edge contour feature, and the number is the number of its corresponding feature. (**a**) Results of edge structure decomposition. (**b**) Result after removing overlapping or similar edges. (**c**) Results of edge fitting and merging.

### 3.2.2. Local Edge Contour Merging

Fitting and reassembling are applied to the decomposed local edge objects, aiming to restore the original and relatively large contours that conform to quadratic curves as much as possible. The two local edge objects to be fitted are merged, forming an input sample for data merging. According to the characteristics of the desired contour structure, this paper selects the quadratic fitting model among the polynomial fitting models. The least squares method is used to fit the point sets of two local edge objects so that the polynomial curve best approximates the given data points, and then the minimized residual sum of squares of the two local edge objects is obtained. A fitting threshold is set, and the two local edge objects that meet the fitting threshold are merged, obtaining the desired local edge contour features. As shown in Figure 7c, local edge Object 1 in c is the result of fitting local edge Objects 1 and 3 in b. If the minimized residual sum of squares is less than the set fitting threshold, the current two local edge objects are considered to have a better fitting effect and are merged into a larger contour structure that conforms to the quadratic curve. Otherwise, the two local edge objects are considered to have no relationship. Finally, the complete desired local edge contour features are obtained.

### 3.3. Local Edge Contour Feature Matching

When the image pair involves perspective distortion, relying solely on affine transformations of translation and rotation to achieve local edge contour feature matching of curve shapes is no longer applicable, because the perspective transformation introduces more complex deformations. In order to solve this problem, this paper calculates the global homography matrix through the matching feature point pairs extracted by SIFT and point-line consistency and performs coarse position matching on the edge contour features of the image overlapping area. Before calculating the global homography matrix, the more robust Multi-GS algorithm [32] is used to filter outliers. In images containing multiple coplanar structures, the Multi-GS algorithm reduces the time complexity of hypothesis model generation, reduces the risk of mismatching, and better adapts to the complexity of the scene. We assume $I$ and $I'$ are the reference image and the target image, respectively. There is a pair of matching points $(x_i, y_i) \in I$ and $(x'_i, y'_i) \in I'$. There is a homography transformation relationship between them, which can be expressed as the following homogeneous linear relationship:

$$\begin{bmatrix} x_i & y_i & 1 & 0 & 0 & 0 & -x'_i x_i & -x'_i y_i & -x'_i \\ 0 & 0 & 0 & x_i & y_i & 1 & -y'_i x_i & -y'_i y_i & -y'_i \end{bmatrix} h = 0. \tag{4}$$

The global homography transformation matrix $h$ can be calculated based on the feature matching point pairs and their linear relationships. The reference image and the target

image are projected into the same coordinate system using the homography transformation *h* to achieve a rough alignment of the local edge contour features.

The overlapping area mask can be obtained by performing an AND operation on the binary images of the target image and the reference image. The overlapping region mask is employed to extract local edge objects within the overlapping region, facilitating feature matching and enhancing the efficiency of feature matching. Extraction rules: The minimum bounding rectangle for each edge is calculated. If the rectangle is entirely located within the overlapping region, all edge data are retained. In cases where the rectangle intersects with the overlapping region, only the edge data corresponding to the intersection are preserved; otherwise, they are disregarded.

During the matching process, edges that are too dense interfere with the matching results of local edge contour features. By performing close-range filtering on local edge objects in the overlapping area, local edge objects that are close and easy to interfere with each other are excluded, while local edge objects with more significant characteristics are retained. First, we assume that the local edge object set $E = \{e_i\}_{i=1,2...n}$ in the reference image is sorted according to the length of the main axis, $E$ represents the set of local edge objects, and $e_i$ represents the local edge object extracted from the reference image. A distance matrix is established, where both rows and columns represent sequentially arranged local edge objects. The values at corresponding positions indicate the distances between the two edges. Then, the distance between each local edge object and other local edge objects is calculated sequentially, as shown in Formula (3), and the distance value is stored in the distance matrix. A minimum edge distance threshold T is defined to assess the distance relationship between two local edge objects. If the distance value is greater than T, it is considered that the two local edge objects do not affect each other, and the corresponding position of the distance matrix is set to 0. Otherwise, the two local edge objects are considered to interfere with each other, and the corresponding position of the distance matrix is set to 1. According to the distance relationship, each $e_i$ in the local edge object set is screened in order to determine whether the object has other local edge objects that interfere with it. If interfering adjacent objects are found, the local edge objects that interfere with $e_i$ are found through the distance matrix. The discovered local edge objects are assigned a value of 0 at the corresponding position in the distance matrix with respect to $e_i$. This implies the elimination of all interference relationships between local edge objects and $e_i$. Simultaneously, local edge object $e_i$ is removed. Upon completing the traversal of the collection of local edge objects, local edge objects free from any interference relationships are obtained from the final distance matrix (where the values in the corresponding rows are all 0). Through the above filtering, a group of local edge objects that maintain a certain distance and are relatively independent are obtained. Local edge objects in the target image also need to undergo the same filtering operation mentioned above.

After the images to be spliced are roughly aligned, the positional relationship of their four vertices is calculated based on the minimum circumscribed rectangle of the local edge object. After undergoing coarse alignment, the positions of the four vertices of the minimum bounding rectangle for the local edge objects are calculated. This ensures that the distances between matched local edge objects remain within the matching error range, thereby obtaining matched pairs of local edge contour features.

We assume that the minimum circumscribed rectangles of the local edge objects in the target image and the reference image are $R_{target} = \{Rect_i\}_{i=1,2,...t}$ and $R_{refer} = \{Rect_j\}_{j=1,2,...r}$, respectively. It is necessary to create relationship matrix $Matrix_{t \times (r+1)}$ surrounding the rectangle of the local edge object. The first column of relationship matrix $Matrix_{t \times (r+1)}$ stores local edge object $Rect_j$ corresponding to local edge object $Rect_i$ with the smallest distance. For each local edge object $Rect_i$ in the target image and each local edge object $Rect_j$ in the reference image, the distance between the four vertices of their circumscribing rectangles is recorded in the relationship matrix. Assuming rectangular vertex coordinates $v_i^k \in Rect_i$ and $v_j^k \in Rect_j$,

the vertex distance of the circumscribing rectangles of two local edge objects is expressed as follows:

$$\text{dist}\left(\text{Re}ct_i, \text{Re}ct_j\right) = \sum \left\| v_i^k - v_j^k \right\|, k = 1, 2, 3, 4. \tag{5}$$

In the formula, $\text{dist}\left(\text{Re}ct_i, \text{Re}ct_j\right)$ represents the sum of the distances of the four vertices of the two circumscribed rectangles, and $v_i^k$ and $v_j^k$ are the vertex representations of the two circumscribed rectangles.

The relationships between local edge contour features are determined through the relationship matrix. However, due to the uncertain quantity of local edge contour features in the image, errors in matching, such as one-to-many or many-to-one, may arise when associating corresponding local edge contour features. To eliminate erroneous matches, the set of local edge contour features with fewer edges in the image is initially chosen for active traversal, while the other set of contour features serves as passive traversal. This approach attributes the aforementioned issue to the potential occurrence of a one-to-many problem. In the one-to-many situation, the distance sum of circumscribed rectangular vertices of multiple matching contour feature pairs that match the same local edge contour feature is calculated and compared, thereby selecting the best matching local edge contour feature. If the local edge contour feature is not the best match, then the local edge contour feature with the smallest distance is selected again in the feature set. Iteration continues until matching local edge contour features are found for all target edge contour features. Finally, the local edge contour features that are empty in the relationship matrix represent local edge contour features that have no corresponding matching.

The above method is only suitable for overall matching of local edge contour features. However, for the matching situation where only intersection data exist for local edge contour features, the method of calculating the vertex distance of the minimum circumscribed rectangle of the entire edge contour is obviously no longer applicable. Therefore, the intersection data of the circumscribed rectangles of local edge contours are used to compare the relationship between edge contours. First, the intersection data of the local edge contours to be compared are obtained, and the corresponding minimum circumscribed rectangle is calculated. If there is an intersection between two edge contours, the distance relationship between the vertices of the circumscribing rectangle is used to determine the matching edge contour; otherwise, it is directly considered that there is no relationship between them.

In addition, multiple factors are also taken into account to consider the matching similarity of local edge contour features. First, the consideration involves the shape similarity of local edge contour features. The method of quadratic fitting is utilized to abstract the shape of local edge contour features into a mathematical model, aiming to determine whether they belong to the same structural type. That is to say, if the shapes of the matched local edge contour features are not all straight lines or curves, the two local edge contour features are considered to be mismatched. Next is the consideration of the positional relationship of local edge contour features. The starting and ending points of local edge contour features are connected to form a straight line, and the similarity between two local edge contour features is measured by the slope of the line. That is to say, the direction sign of the slope of the connecting straight line is calculated. If the sign of the straight line slope of the two local edge contour features is different, the two local edge contour features are considered to be mismatched. This results in more accurate pairs of matched local edge contour features.

The results obtained through the process of constructing and matching edge contours on multiple images with overlapping regions are depicted in Figure 8.

### 3.4. Image Pre-Alignment

Matching point $p = \{p_i\}_{i=1}^N$ added by point-line consistency [12] and matching feature point $p' = \{p_i'\}_i^N$ extracted by SIFT are merged into matching point pair data $P = \{(p_i, p_i')\}_i^N$, which represents the input N sets of data. Multi-GS [26] is used to filter interior points, and M fitting model parameter sets $\{\theta_1, \theta_2, \ldots, \theta_M\}$ are generated by random sampling. We assume a local edge contour feature pair $\left(e_j, e_j'\right)$, where j represents

the identifiers for the local edge contour feature pair. For each hypothetical model, first, we calculate distance $D\left(e_j, e'_j\right)$ between the two transformed local edge contour feature pairs, which is obtained by Formula (3). The required edge distance is also the error value of the local edge contour feature pair under this model. Second, distance error sum $S_i = \sum D\left(e_j, e'_j\right)$ of all local edge contour feature pairs is calculated. The distance error sums of local edge contour feature pairs under all model assumptions are sorted from small to large to obtain an ordered list $S_{sorted}$. Third, mean square error $MSE_j$ is calculated for the sum of error distances for all pairs of local edge contour features in a single model hypothesis. Next, it is necessary to calculate mean error value $u_j$ of all local edge contour feature pairs, and sort the obtained results from small to large. This is to rule out hypothetical models in which the overall alignment is affected because a single local edge contour feature is severely misaligned. The formula is expressed as

$$MSE_i = (1/n) * \sum \left(D\left(e_j, e'_j\right) - u_j\right)^2. \tag{6}$$

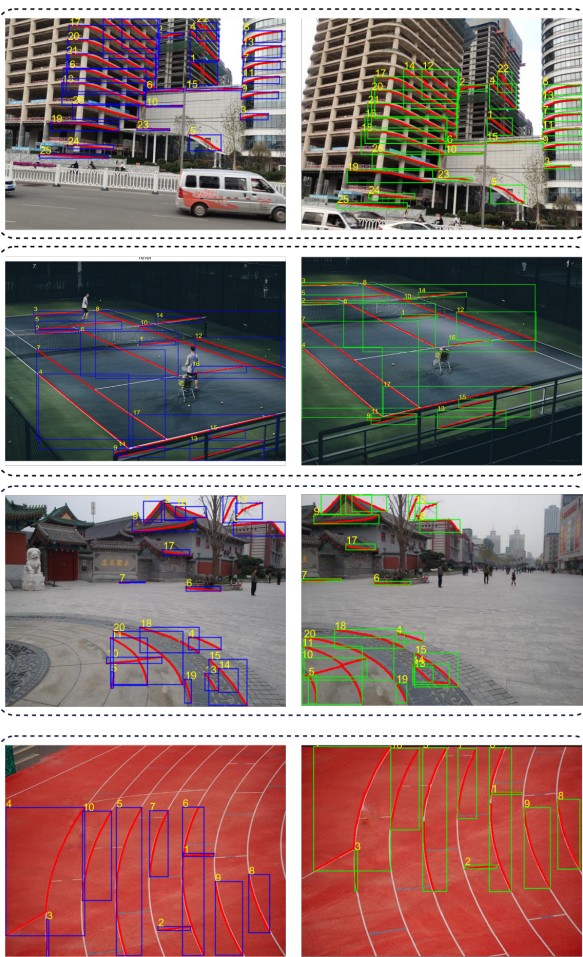

**Figure 8.** The results of local edge contour matching across multiple datasets. The blue and green boxes are the detected local edge contour features. Each pair of features has the same number.

In the formula, n represents the number of edge contour pairs, and $u_j$ represents the distance variance of all edge matching pairs under the assumption of model $\theta_i$. Then, the calculated variance values are sorted to obtain an ordered list, $MSE_{sorted}$. According to the two ordered lists, $S_{sorted}$ and $MSE_{sorted}$, the hypothesis model ranked highest in both lists is the best hypothesis model sought.

### 3.5. Grid Optimization with Multiple Constraint Terms for Warping

The previously discussed global pre-alignment, which relies on local edge contour constraints, is mainly used to ensure alignment of the entire image. To meet the demands of local alignment in perspective-transformed images and to mitigate issues such as artifacts and distortions, the image is divided into a dense grid, and local pre-warping is applied within each grid. An energy function is formulated for grid vertices [12], incorporating a range of prior constraint terms, to guide the deformation of the grid through energy minimization. This paper defines the grid vertex optimization energy function from the following four aspects.

(1) Local point alignment constraint term

A feature point alignment term is designed to ensure that the matched feature points remain aligned during the grid deformation process. For target image $I$, reference image $I'$, and extracted feature points $p_i \in I$ and $p'_i \in I'$, this constraint term is defined as follows:

$$E_p = \sum_{i=1}^{N} \left\| \tau(\widehat{p}) - p'_i \right\|^2 = \left\| W_p \widehat{V} - P \right\|^2, \tag{7}$$

where N represents the quantity of matching points, $\tau(\widehat{p})$ denotes the positions of points $p_i$ after transformation, which are interpolated by the grid vertices, and $W_p$ represents the weight matrix corresponding to point set $\{p_i\}_{i=1,2,\dots,N}$.

(2) Local line alignment term

The local line alignment option not only further optimizes the local grid alignment effect, but also makes the linear alignment effect in overlapping areas more stable. The alignment of matching straight lines is constrained by the shortest distance between the end points of the straight line segment on the target image and the matching straight line in the reference image. According to the matching straight line cut off by different grids, the short line segment formed intersects with the grid. This paper enhances the constraints on the alignment of the straight line through the shortest distance from the intersection point to the matching straight line. For the collection of matching line pairs denoted as $\left\{ l_j, l'_j \right\}_{j=1,2,\dots,M}$, and for their corresponding matches $l_j \in I$ and $l'_j \in I'$, the constraint term is defined by the following formula:

$$E_l = \sum_{j=1}^{M} \left\| \frac{l'^T_j \cdot \tau(p_i^{s,e})}{\sqrt{a_i^2 + b_i^2}} \right\| = \left\| W_l \widehat{V} \right\|. \tag{8}$$

In the equation, $p_i^{s,e}$ represents the two endpoints of the straight line segment and the intersection point of the two ends of the truncated short straight line. $l'^T_j = (a_i, b_i, c_i)$ are coefficient vectors for the linear equations of line segments in the reference image. Utilizing the correspondence between points and lines, the local line alignment term involves optimization to determine the minimum distance from the two endpoints to the line.

(3) Long line constraint term

This constraint is based on the LCP [12], which merges local straight line segments on the same collinear structure of the image into long straight lines to protect the long straight line structure in the image from bending. For each long straight-line structure $L_{gl} = \{l_i\}_{i=1,2,\dots,s}$, uniform sampling is conducted along each long line, represented as $\left\{ p_i^k \right\}_{k=1,2,\dots,Q}$. This constraint term is expressed by the following formula:

$$E_{gl} = \sum_{i=1}^{S} \sum_{k=1}^{Q} \left\| \frac{l'^T_i \cdot \tau\left(p_i^k\right)}{\sqrt{a_i^2 + b_i^2}} \right\| = \left\| W_{gl} \widehat{V} \right\|. \tag{9}$$

In the equation, $l_i'^T = (a_i, b_i, c_i)$ represents the coefficient vector of the linear equations constructed from the two endpoints of long line segments $(u_i^s, v_i^s)$. The combination of local short line and global long line constraint terms ensures that the line structures in the image maintain better results both locally and overall.

(4) Global constraint term

Sets of horizontal lines, denoted as $\left\{ \left( l_i^u, l_i'^u \right) \right\}_{i=1}^S$, and vertical lines, denoted as $\left\{ \left( l_i^v, l_i'^v \right) \right\}_{j=1}^T$, are generated from the registered image. Uniformly sample points for each horizontal and vertical line form point sets $\left\{ p_k^{u,i} \right\}_{k=1,2,\dots,L_1}$ and $\left\{ p_k^{v,i} \right\}_{k=1,2,\dots,L_2}$, respectively. The global image distortion is constrained by the slopes of horizontal and vertical lines, as expressed in the following formula:

$$
\begin{aligned}
E_{gs} = &\sum_{i=1}^S \sum_{k=1}^{L_1} \left\| \frac{l_i'^T \cdot \tau\left(p_k^{u,i}\right)}{\sqrt{a_i^2 + b_i^2}} \right\| \\
&+ \sum_{j=1}^T \sum_{k=1}^{L_2} \left\| \frac{l_j'^T \cdot \tau\left(p_k^{v,i}\right)}{\sqrt{a_j^2 + b_j^2}} \right\| \\
&+ \sum_{j=1}^T \sum_{k=1}^{L_1-2} \left\| \tau\left(p_k^{v,j}\right) + \tau\left(p_{k+2}^{v,j}\right) - 2\tau\left(p_{k+1}^{v,j}\right) \right\|.
\end{aligned}
\tag{10}
$$

Additionally, to constrain deformation in non-overlapping regions, the constraint term is expressed as follows:

$$
E_{gd} = \sum_{i=1}^S \sum_{k=1}^{\left\lfloor p_k^{v,j} \in \Phi \right\rfloor - 2} \left\| \tau\left(p_k^{u,i}\right) + \tau\left(p_{k+2}^{u,i}\right) - 2\tau\left(p_{k+1}^{u,i}\right) \right\|^2 = \left\| W_d \hat{V} \right\|^2.
\tag{11}
$$

Therefore, the global constraint term defined in this section is expressed as follows:

$$
E_g = \lambda_{gs} E_{gs} + \lambda_{gd} E_{gd}.
\tag{12}
$$

$\lambda_{gs}$ and $\lambda_{gd}$ represent the weights for constraint terms of the line slope and the non-overlapping region, respectively.

The four constraint terms for the energy function described above can be combined into a unified energy function optimization problem, as shown below:

$$
E = \alpha E_p + \beta E_l + \gamma E_{gl} + E_g.
\tag{13}
$$

In the equation, $\alpha$, $\beta$, and $\gamma$ correspond to the weights of the energy function constraint terms. Since each term is quadratic, a sparse linear solver is employed to minimize the energy function and solve the grid optimization problem outlined above.

### 3.6. Linearly Weighted Image Fusion

Image fusion is the process of reassembling images based on their completed registration positions and orientations. Exceptional image fusion algorithms can reduce and in some cases eliminate issues related to brightness, color, and stitching seam discrepancies in overlapping regions, ensuring smoother transitions between overlapping and non-overlapping areas. In this section, we provide a detailed explanation of the linear weighted fusion algorithm [38] employed in the paper.

We assume that reference image $I_1$ and target image $I_2$ are linearly weighted fused. We calculate the weight based on the distance from the overlapping part to the boundary of the overlapping area, and normalize the weight. We use $w_1$ and $w_2$ to represent the weight, then $w_1 + w_2 = 1$. A linear weighted sum is performed for each position, and the summation formula is as follows:

$$F(x,y) = w_1 \cdot I_1(x.y) + w_2 \cdot I_2(x.y). \tag{14}$$

In the formula, $F(x,y)$ represents the pixel value of the fused image at position $(x,y)$.

In order to ensure that the pixel value of the fused image is within a reasonable range, normalization processing is performed as shown in the following formula:

$$F(x,y) = \frac{F(x,y)}{w_1 + w_2}. \tag{15}$$

In this way, through pixel-by-pixel linear weighted summation, fused image F is obtained.

## 4. Experimental Design and Result Analysis

### 4.1. Experimental Preparation

4.1.1. Experimental Environment and Setup

In order to verify the effectiveness of the proposed image mosaic algorithm based on point-line consistency and local edge contour constraint, detailed experimental tests and comparative analysis were carried out in this chapter to evaluate the performance of the algorithm. Several classical image mosaic algorithms, including SIFT+RANSAC, APAP [15], AANAP [28], SPW [22], and LCP [12], were selected and compared with representative image datasets. In the experiments, the input image size was consistently set to $640 \times 480$ pixels. During the edge contour extraction, the Canny detection algorithm utilized low and high thresholds of 0.001 and 0.2, respectively. In addition, the point distance threshold was 3, the principal axis length threshold was 50, the edge similarity threshold was 15, the fitting threshold was 4, and the minimum edge distance threshold was 80. In the grid optimization process, the grid pixel size was set to $40 \times 40$. Additionally, the weight values for stages $\alpha$, $\beta$, $\gamma$, and $\lambda_{gs}$, $\lambda_{gd}$ during the energy function optimization were assigned as follows based on prior literature and practical testing results: 1, 5, 100, 50 and 100. The algorithm was realized by MATLAB, part of which was programmed by C++. The experimental environment was configured as Intel(R) Core(TM)i5-6300HQ CPU and 16 GB RAM.

4.1.2. Evaluation Metrics

This section effectively evaluates image stitching from two aspects, visual evaluation and objective index evaluation. Visual evaluation is mainly based on the subjective feeling of the human eye and evaluates the stitched image. The following aspects are usually considered: 1. The overall appearance of the stitched image and whether there are problems such as deformation, stretching and distortion; 2. Examination of whether the overlapping area is accurately aligned with pixels to ensure natural transition; 3. Examination of the integrity of the structural information and whether significant structural information such as shapes and lines in the original image is damaged. Because of the complex scene of image stitching design, it is difficult to measure its quality by objective evaluation model in many cases. Therefore, visual evaluation is still a very important way to evaluate the quality of image stitching.

Root mean square error (RMSE) and structural similarity index (SSIM) [39] are mainly used in the common quality evaluation indexes of stitched images. RMSE is used to measure the difference and transformation degree of data, especially the matching effect of matching points. It measures the error by comparing the original position of feature points with the position after splicing. A smaller RMSE indicates a better alignment effect of feature points. The formula is as follows:

$$RMSE(t) = \sqrt{\frac{1}{N} \sum_{i=1}^{N} \left\| \tau(p_i) - p_i' \right\|^2}. \tag{16}$$

In Equation (16), $p_i$ represents the position of feature points in the target image, $p'_i$ represents the position of feature points in the reference image, $N$ stands for the number of feature point pairs, and $t$ represents the transformation between the stitched images.

SSIM is primarily utilized to assess the similarity of the overlapped regions between two stitched images in terms of contrast, illumination, and structure. It combines luminance, contrast, and structure to yield a unified similarity score, which is employed to evaluate the likeness of the overlapped area. The calculation formula for this parameter is as follows:

$$SSIM(x,y) = \frac{\left(2u_x u_y + C_1\right) + \left(2\sigma_{xy} + C_2\right)}{\left(\mu_x^2 + \mu_y^2 + C_1\right)\left(\sigma_x^2 + \sigma_y^2 + C_2\right)}. \tag{17}$$

In Equation (17), $\mu_x$ and $\mu_y$ correspond to the mean pixel intensities of the overlapped regions in the target and reference images, respectively. $\sigma_x$ and $\sigma_y$ represent the standard deviations of pixel intensities, $\sigma_{xy}$ stands for covariance, and $C_1$ and $C_2$ are constants introduced to prevent division by zero. A higher SSIM value indicates a greater similarity between the overlapped regions of the target and reference images. When the SSIM value is 1, it signifies that the pixel values of the two images are perfectly identical.

### 4.2. Experimental Results and Analysis

### 4.2.1. Experiments on Feature Augmentation Using Point-Line Consistency

The main purpose of this experiment is to verify the effect of point-line consistency method on increasing the number of feature matching. In the experiment, SIFT descriptors were chosen for feature extraction and matching on adjacent images. Subsequently, Mutil-GS algorithm is used to filter out the wrong matching points to obtain effective matching inner point pairs. At the same time, the block strategy of point-line consistency is adopted to extract and match the features of the local area of the image in order to obtain more rich and effective matching features. Table 1 shows the results of feature extraction and matching between the two feature matching methods on multiple datasets.

**Table 1.** The comparison results between our method and the SIFT method.

| | APAP-Conssite | DHW-Carpark | Office | Sportfield1 | Laodong Park | Rail Station | Olympic Building |
|---|---|---|---|---|---|---|---|
| SIFT+RANSAC | 229/390 | 287/391 | 89/219 | 138/217 | 183/273 | 330/516 | 253/380 |
| OUR | 814/833 | 2266/2742 | 222/271 | 653/810 | 444/517 | 1523/1727 | 2078/2490 |
| Increase Ratio | 255.46% | 689.55% | 149.44% | 373.19% | 142.62% | 361.52% | 721.34% |

The data in Table 1 represent the ratio of the number of matching points obtained by the two different feature extraction methods to the number of interior points filtered by the Mutil-GS algorithm, as well as the proportion of the method used in this article to increase the number of interior points compared to the SIFT method. As the number of corresponding features increases, the information entropy of the image also increases, containing more details and structural information, making it more accurate and clear. It can be clearly seen from the data in the above table that the point-line consistency method used in this article can obtain more rich and effective feature matching points, which helps to achieve better results in image registration tasks.

### 4.2.2. Contour Feature Ablation Experiment

In this module, the extracted local edge features are used in the global pre-registration phase as contour matching pairs to restrict the global better alignment of the image, and the ablation experiment is carried out. Disruption experiments are conducted to validate its effectiveness. In this ablation experiment, we focus on comparing two different image mosaic methods, the image mosaic method using point-line consistent matching pair for global preregistration, and the image mosaic method based on point-line consistent matching pair and local edge contour feature constraints. The results of the two sets of

experiments are compared in detail to see their differences and to verify the effect and effectiveness of the constraint of local edge contour features in the stitching algorithm.

Figure 9 is the visual comparison of the results of the ablation experiments of the two contrast experiments. The stitched images are obtained from the classic datasets, APAP-railtracks and RailStation, respectively. It is found that the stitching method based on point-line consistency and contour feature constraint can achieve precise registration and alignment better than the point-line feature matching method based on local block point-line consistency constraint extraction. In Figure 9a,b, the left figure shows the result of stitching based on point-line consistency, and the right figure shows the result of stitching based on point-line consistency and contour feature constraints. In (a), the left figure is not aligned and artifacts occur, while the right image, constrained by contour features, exhibits a significantly improved alignment with no artifacts. In (b), the enlargement of the section within the red box in the left image reveals disruptive noise and artifacts in the stitching result, whereas the right image, incorporating contour feature constraints proposed in this paper, demonstrates a notable enhancement in the alignment of line contours when compared to the left image.

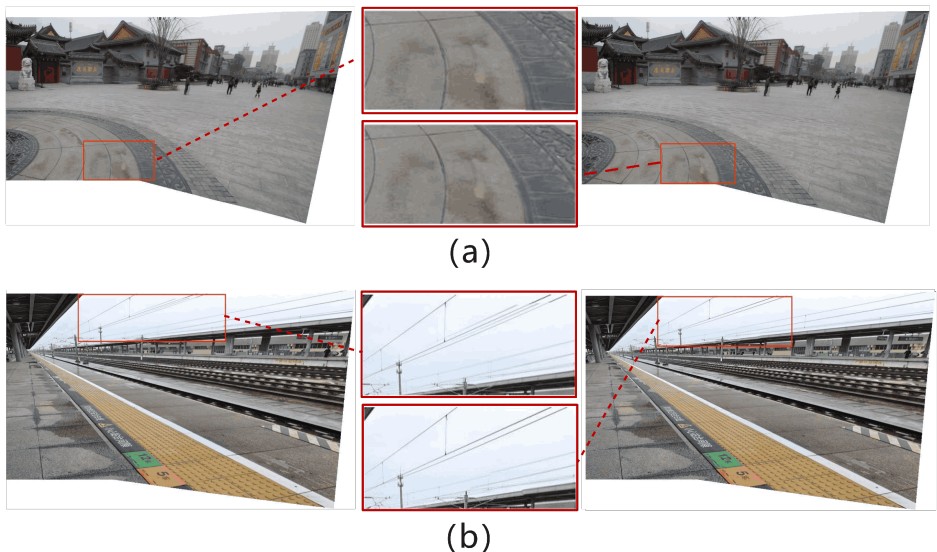

(a)

(b)

**Figure 9.** Results of disruption experiments. The rectangular frame in the middle is an enlargement of the red frame position in the splicing result, and is also the key comparison position in the splicing result.

For the two pairs of image datasets above, (a) and (b), the edge contour construction method proposed in this article is used to extract edge contours and perform contour matching. Then, a fitting operation is performed on the extracted edge contour matching pairs, and the mean sum of squares of the residuals of each contour matching pair is calculated. The results are shown in Tables 2 and 3.

**Table 2.** The sum of squared residuals from profile fitting in dataset (a).

| Constraints | edge_1 | edge_2 | edge_3 | edge_4 | edge_5 | edge_6 | edge_7 | edge_8 | edge_9 | edge_10 | edge_11 | edge_12 |
|---|---|---|---|---|---|---|---|---|---|---|---|---|
| No | 0.9747 | 0.0988 | 0.3746 | 0.2753 | 7.4168 | 2.2124 | 0.2708 | 4.6931 | 0.9707 | 0.1250 | 4.0343 | 0.2225 |
| Yes | 1.2127 | 0.0752 | 0.3148 | 0.2589 | 3.3984 | 2.2122 | 0.2704 | 4.4830 | 1.2702 | 0.1158 | 3.3984 | 0.2283 |

**Table 3.** The sum of squared residuals from profile fitting in dataset (b).

| Constraints | edge_1 | edge_2 | edge_3 | edge_4 | edge_5 | edge_6 | edge_7 | edge_8 | edge_9 | edge_10 | edge_11 | edge_12 |
|---|---|---|---|---|---|---|---|---|---|---|---|---|
| No | 0.1201 | 0.1978 | 0.1724 | 0.0747 | 1.3179 | 1.1969 | 0.0949 | 0.3459 | 3.8874 | 1.6272 | 0.0951 | 0.1342 |
| Yes | 0.0494 | 0.4955 | 0.1541 | 0.0709 | 1.0664 | 0.9058 | 0.0719 | 0.3448 | 4.0613 | 1.3591 | 0.0910 | 0.1244 |

In order to better compare the differences between the two sets of data, a line diagram to display the data more significantly indicates the distribution of the scattered value. The folding line chart is shown in Figure 10. The blue point represents the sum of squared residuals after fitting the edge contour matching pairs in the image stitching result without contour constraints. The red points represent the sum of squared residuals after fitting for edge-contour matching pairs with contour constraints. Observing the data points of the line chart, it can be clearly seen that the blue point data are generally distributed above the red data. Under the constraints of edge contour features, the edge contour in the splicing result has a better fitting effect.

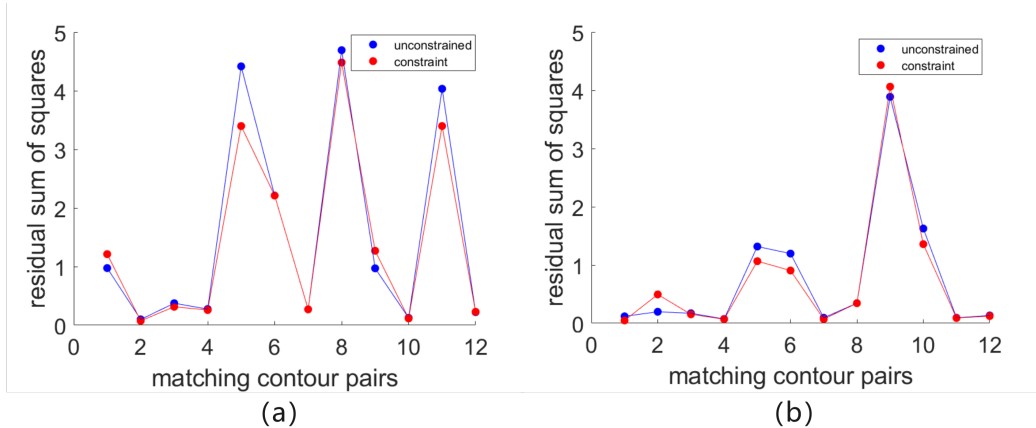

**Figure 10.** Residual sum of squares calculated for contour matching pairs after warping and fitting. The X-axis represents the contour matching pairs, and the Y-axis represents the calculated residual sum of squares. (**a**) is the fitting result of the local edge contour features detected on dataset a in Figure 9. (**b**) is the fitting result of the local edge contour features detected on dataset b in Figure 9.

4.2.3. Algorithm Visual Evaluation

In this experiment, SIFT+RANSAC, APAP, AANAP, SPW, and LCP are selected for visual evaluation. The dataset grail is used. The following image is the test result of the preceding algorithms on the dataset.

Figure 11 is the splicing result of the above algorithms on the dataset grail. The comparison results in Figure 11 highlight the advantages of the splicing method in this paper. Particularly in the prominent contour regions of the left window and the middle wall, several algorithms exhibit noticeable disparities. Compared with other splicing results, the SIFT+RANSAC algorithm has the worst splicing effect, and the artifact problem is very obvious. The stitching results of the APAP and AANAP algorithms reveal that lines at the window locations overlap and fail to align correctly, resulting in significant artifacts. In contrast, the SPW, LCP, and the algorithm proposed in this paper offer notably superior stitching results at the window locations. This improvement is attributed to the introduction of local straight-line constraints, which help better confine straight-line features. Moreover, the LCP algorithm introduces extended straight-line constraints, further reducing the occurrence of artifacts. Additionally, for the prominent edges on the middle wall, the stitching results of the SIFT+RANSAC, APAP, AANAP, SPW, and LCP algorithms display evident misalignment, leading to the presence of artifacts. The stitching algorithm proposed in this paper excels in terms of alignment, exhibiting minimal artifacts and yielding clearer and more precise results.

From the effect comparison in Figure 12, it can be observed that the SIFT+RANSAC, APAP, AANAP, SPW and LCP algorithms maintain the original outline of the building in the overall visual feel of the panorama, but their performance in local details is not satisfactory. In particular, there is obvious misalignment and ghosting in the alignment effect on the glass railings, and a certain degree of distortion at the edges. In comparison, the algorithm proposed in this article has almost no obvious ghosting phenomenon in

the splicing results, and the distortion of the railing edges is slighter, making the splicing results clearer.

### 4.2.4. Objective Evaluation of Algorithm Performance

In the experiments, RMSE and SSIM of images were computed using different methods on various datasets. The selected datasets were characterized by easily extractable local edge contours and a substantial number of edges. Especially the first four datasets featured prominent curved edge contours. The RMSE values indicate that, compared to other algorithms, the algorithm proposed in this paper achieves a higher level of alignment for matched feature points after image fusion. The specific results of the root mean square error (RMSE) test are shown in Table 4.

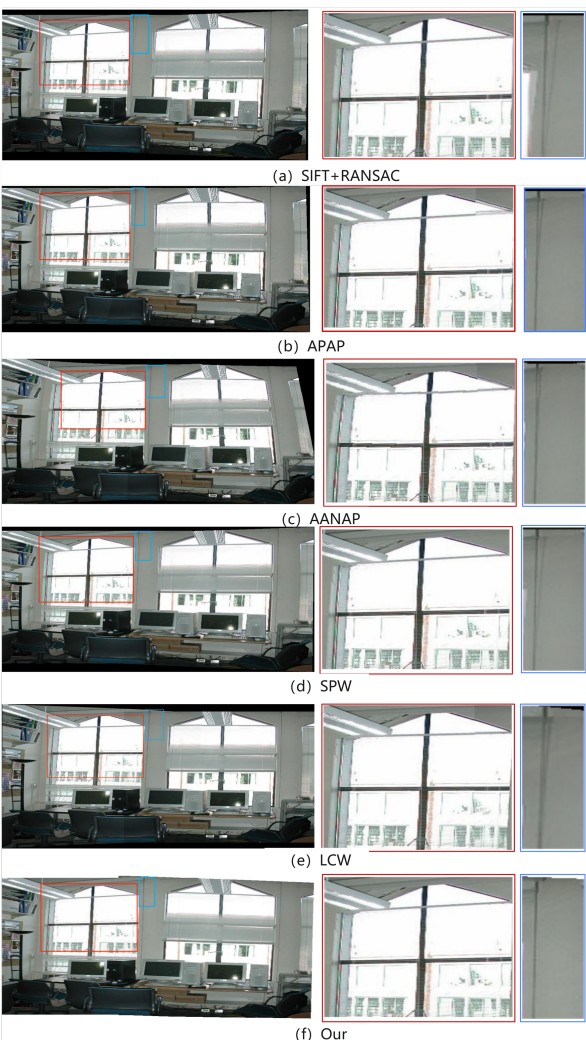

**Figure 11.** The splicing results of each algorithm on the grail dataset. Zoom in on the location of the window in the red box and the prominent edge of the wall in the blue box.

Furthermore, structural similarity comparison was performed on the stitched images to assess the consistency of pixels in the overlapping regions between the target image and the reference image after registration-based deformation. The specific results of the Structural Similarity Index (SSIM) testing are presented in Table 5.

A higher SSIM value, approaching 1, indicates a greater level of alignment between the two images in the overlapping regions after the completion of the stitching process. Upon comparing the SSIM test results, it is evident that, although in the Library and sportfield1 datasets, the data from our algorithm are slightly lower than those of SPW, it is better than

other algorithms in the overall dataset performance. Therefore, the aforementioned data demonstrate that our algorithm achieves a higher level of accuracy in feature matching during the image stitching process on the mentioned datasets.

At the same time, charts are used to intuitively display the distribution of data and comparison results. Figure 13a shows the RMSE data calculated by the above algorithms. It can be clearly seen that the data calculated by our method are overall distributed below, with smaller values. It is shown that the matching feature points in the overlapping areas in the splicing results of our method have better alignment effects. Figure 13b shows the SSIM data calculated by the above methods. It is easy to see that the data calculated by our method are overall distributed above. It is shown that the splicing results of this method have higher structural similarity in the overlapping areas and better alignment effect.

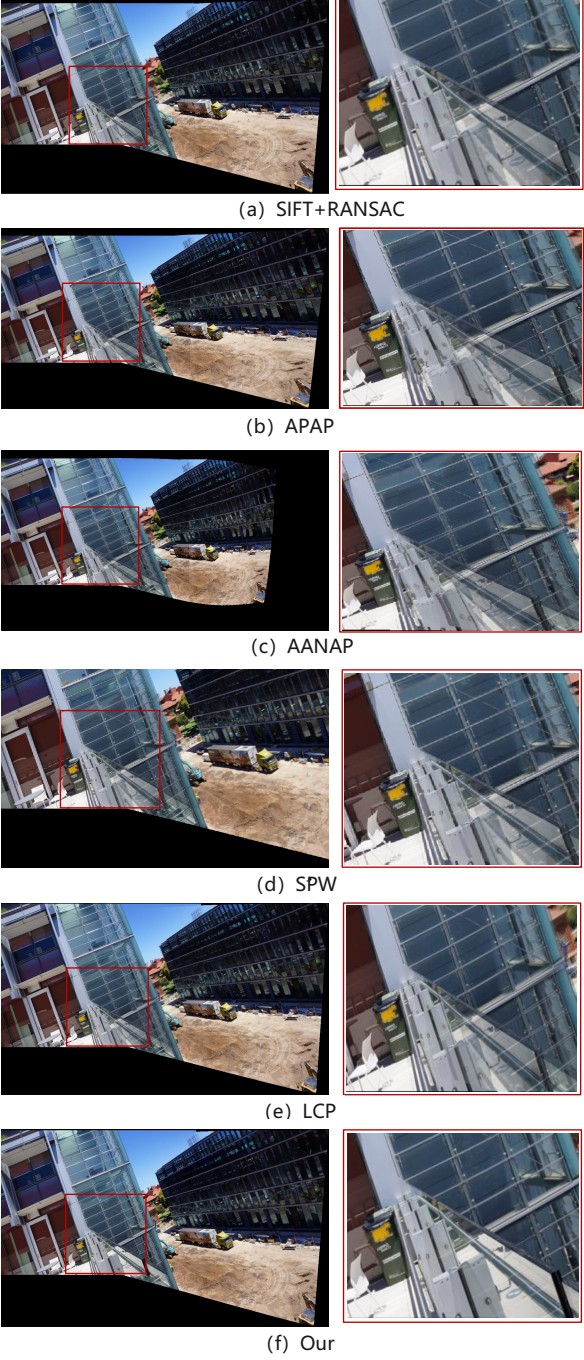

(a) SIFT+RANSAC

(b) APAP

(c) AANAP

(d) SPW

(e) LCP

(f) Our

**Figure 12.** The splicing results of each algorithm on the dataset conssite, and the area in the red box is enlarged.

**Table 4.** The RMSE results of each stitching algorithm on the test dataset.

| Datasets | SIFT + RANSAC | APAP | AANAP | Spw | LCP | Ours |
|---|---|---|---|---|---|---|
| APAP-railtracks | 9.16 | 5.79 | 6.38 | 5.22 | 4.91 | 4.40 |
| GES-Building | 11.66 | 5.81 | 5.18 | 2.11 | 2.07 | 1.66 |
| GES-Garden | 8.05 | 5.63 | 5.39 | 3.43 | 2.90 | 2.82 |
| Library | 7.78 | 5.47 | 5.12 | 3.83 | 2.66 | 2.58 |
| DFW-shelf | 7.67 | 5.93 | 5.62 | 3.8 | 3.94 | 3.73 |
| SPHP-bridge | 4.40 | 3.76 | 3.71 | 1.93 | 2.20 | 1.88 |
| SPHP-building | 6.54 | 5.55 | 5.03 | 3.43 | 3.13 | 2.97 |
| sportfield1 | 7.39 | 5.97 | 5.27 | 4.78 | 4.34 | 4.06 |

**Table 5.** The SSIM results of each stitching algorithm on the test dataset.

| Datasets | SIFT + RANSAC | APAP | AANAP | SPW | LCP | Ours |
|---|---|---|---|---|---|---|
| APAP-railtracks | 0.5372 | 0.5580 | 0.5487 | 0.5691 | 0.5794 | 0.6059 |
| GES-Building | 0.4959 | 0.5760 | 0.5262 | 0.6192 | 0.6554 | 0.6735 |
| GES-Garden | 0.6671 | 0.6973 | 0.7040 | 0.7065 | 0.7533 | 0.7693 |
| Library | 0.6346 | 0.7585 | 0.7592 | 0.8646 | 0.7987 | 0.8307 |
| DFW-shelf | 0.7109 | 0.8119 | 0.8033 | 0.8575 | 0.8409 | 0.8448 |
| SPHP-bridge | 0.5851 | 0.5959 | 0.5649 | 0.6004 | 0.5621 | 0.6139 |
| SPHP-building | 0.5671 | 0.6525 | 0.6091 | 0.6442 | 0.6681 | 0.6708 |
| sportfield1 | 0.7204 | 0.7565 | 0.7686 | 0.8056 | 0.7811 | 0.7978 |

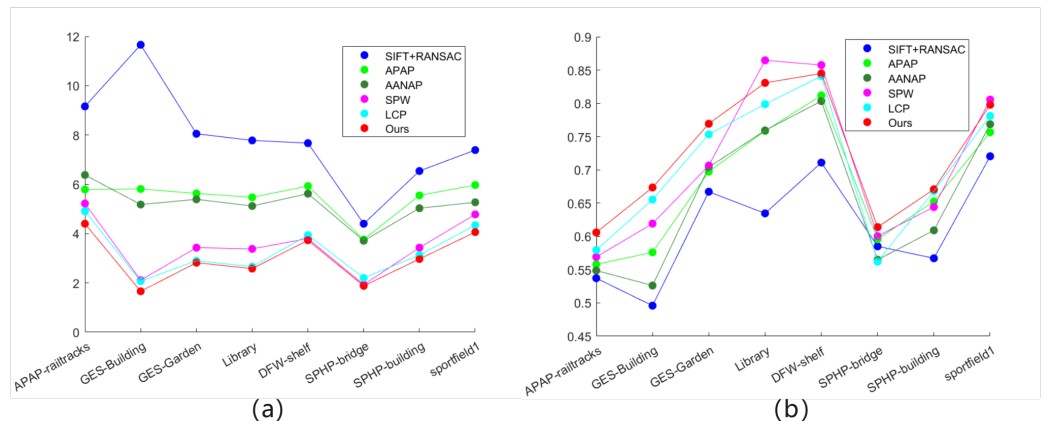

**Figure 13.** (**a**) RMSE numerical display. (**b**) SSIM numerical display, where Ours is the result of the algorithm proposed in this article.

4.2.5. Run Time Analysis

In practical applications, image stitching systems need to have efficient running speed, especially for real-time or interactive application scenarios. Therefore, the experiments in this section conduct an in-depth analysis of the running time of the above-mentioned image stitching algorithms to evaluate their performance in processing complex scenes. Figure 14 shows a comparison of the running times of the above-mentioned algorithms for splicing different datasets.

In Figure 14, it can be easily seen that the running time of several algorithms in different scenarios is very different. The sift+ransac, anap and aanap algorithms are relatively simple, have short running times, and are suitable for fast splicing in simple scenes. The SPW algorithm shows a long execution time when running, showing obvious time-consuming problems. Although the algorithm in this paper has a shorter running time than the SPW algorithm and is similar to the running time of the LCP algorithm, the method in this paper is still not ideal in real-time splicing tasks.

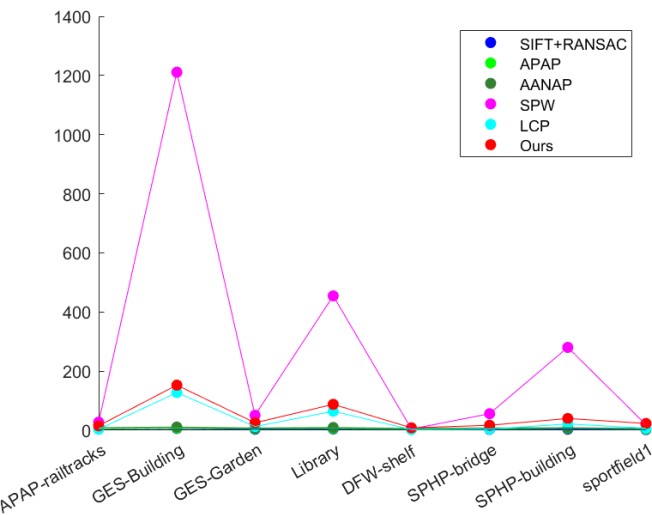

**Figure 14.** Comparison of the running time of the splicing algorithm. The X-axis represents the spliced dataset, and the Y-axis represents the running time.

## 5. Conclusions

This paper introduces a novel image mosaic method which aims to improve the accuracy of image mosaic and solve the artifact problem caused by perspective transformation. Image stitching holds significant relevance in practical applications as it seeks to synthesize images with a broader field of view and richer information. However, the feature matching problem usually leads to mis-estimation of thei mage registration model, which degrades the quality of stitching.

In order to tackle this issue, this paper proposes an image stitching method based on point-line consistency and local edge contour constraints. First, the method uses geometric invariance to increase the number of feature matching points to enrich the matching information. Based on the results of Canny edge detection, significant local edge features are constructed by means of structure separation and edge contour merging to enhance the effect of image registration. The similarity of edge-contour pairs is evaluated by comprehensively considering multiple factors such as shape similarity and edge position relationship. Simultaneously, this paper introduces a spatial transformation warping method to ensure local alignment in the overlapping regions. The constraint of short and long lines is used to maintain the straight line structure in the image to avoid its bending, and the distortion of the non-overlapping area is eliminated by the global line guided warpage. Through comparative experiments and result analysis, the method in this article performs well on multiple datasets and achieves excellent image splicing effects.

However, the experimental results of the running time show that the running time of the splicing algorithm is significantly affected by the complexity of the scene. When processing scenes with a large amount of details or textures and large changes, the algorithm requires more computing time to match feature points or perform splicing operations, thus affecting real-time performance. At the same time, in splicing scenarios dominated by point or straight line features, the edge contour constrained registration effect in the algorithm proposed in this article is slightly inferior to the registration effect of point-line combination. This is because the distribution and importance of different types of features in the scene are different, resulting in a certain imbalance in the matching effects of different types of features. Therefore, future research can be devoted to optimizing the real-time performance of the splicing algorithm, adjusting feature matching and constraint strategies according to changes in the scene, better adapting to splicing requirements under different times and feature distributions, and improving the flexibility of the algorithm.

**Author Contributions:** Conceptualization, S.M. and X.L.; methodology, S.M.; software, S.M.; validation, S.M. and K.L.; formal analysis, S.M.; investigation, S.M.; resources, S.M.; data curation, S.M. and T.Q.; writing—original draft preparation, S.M.; writing—review and editing, S.M.; visualization, S.M. and Y.L.; supervision, X.L.; project administration, X.L.; funding acquisition, X.L. All authors have read and agreed to the published version of the manuscript.

**Funding:** This research was funded by the National Natural Science Foundation of China under grant numbers 61433012, the Natural Science Foundation of Xinjiang Province under grant number 2020D01C026, and the National Natural Science Foundation of China under grant numbers 62261053.

**Institutional Review Board Statement:** Not applicable.

**Informed Consent Statement:** Not applicable.

**Data Availability Statement:** The data in this study are available on request from the corresponding author.

**Acknowledgments:** The authors would like to thank the anonymous reviewers for their valuable comments and suggestions, which helped improve this paper greatly.

**Conflicts of Interest:** The authors declare no conflict of interest.

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
