# Peer review of "Research on Image Stitching Algorithm Based on Point-Line Consistency and Local Edge Feature Constraints"

_entropy, doi:10.3390/e26010061_

Round 1

Reviewer 1 Report

Comments and Suggestions for Authors

The authors propose an image stitching method in which the key innovation is the use of arbitrary image contours for feature-based image alignment. This would appear to be a promising enhancement to well-studied point-and-line feature matching approaches.

The Introduction and Related Work do a fine job of setting up the problem to be solved and describing previous work in the field, but the Materials and Methods section is completely unsatisfactory. Section 3.1 begins by rehashing the work of Jia et al. on coplanar line-point invariants (references 12 and 27) but omits so much material that the variables and equations that are included make no sense. This section only needs to explain how the proposed approach makes use of coplanar line-point invariants, not define it from first principles. In sections 3.2 and 3.3, standard algorithms from image processing (such as the Canny filter and morphological dilation) are needlessly described at length when they could be instead referenced by name and cited. The rest of the text in sections 3.2 and 3.3 (line 267-424), which describes the actual novel elements of the authors’ edge contour feature generation and alignment algorithms, is too unclear to follow and needs to be rewritten entirely. This is the most concerning weakness in the manuscript by far. Section 3.4 rehashes the whole Multi-GS algorithm (reference 26) and should be replaced by more minimal text stating how Multi-GS was used. The grid warping approach in Section 3.5 appears very similar to Section 4 of Jia 2021 (reference 12); please reference this paper again here and explain the similarities and differences to that work.

Section 4 provides an adequate evaluation of the proposed method against other published algorithms. The authors thoughtfully explain the various parameter values chosen for the experiments. The photographic images are too low-resolution to be able to see anything of interest, but I assume this is just an issue with the review draft. The points in Figure 10 don’t seem to bear any relationship to the numbers in Tables 2 and 3; was the wrong figure inserted? It would be helpful to provide plots for the results in Tables 4 and 5 as well. The authors must describe the weaknesses and limitations of their algorithm.

Comments on the Quality of English Language

Minor grammar issues, but overall the English was very understandable.

Reviewer 2 Report

Comments and Suggestions for Authors

This work proposes a new image stitching method based on the point-line consistency and local edge feature constraints. In particular, the point-line consistency module is proposed to increase the number of matching point pairs and filter out erroneous matches. Moreover, this work introduces multiple optimization modules to ensure image alignment and minimize the distortion of non-overlapping regions. Experimental results demonstrate the superiority of this work. Overall, the motivation of this work is sound, and the paper is easy to follow. However, I have some concerns as follows.

- The comparison methods lack some classical image stitching methods, such as “SIFT + RANSAC” and “Robust ELA”.

- It would be interesting to show the comparisons on running/inference time of each method.

- Some important related works of image stitching are missing. For example, "Unsupervised deep image stitching: Reconstructing stitched features to images", "A view-free image stitching network based on global homography", "Pixel-Wise Warping for Deep Image Stitching", "Learning Pixel-wise Alignment for Unsupervised Image Stitching", "Implicit Neural Image Stitching With Enhanced and Blended Feature Reconstruction", etc. The authors are suggested to briefly review the above literature and discuss the development relationship between the learning-based image stitching methods and traditional vision image stitching methods.

- The novelty of the linearly weighted image fusion seems to be limited since a widely known progressive fading method is directly applied. More clarifications about how it works in your scenes are expected to be provided.

- All figures in the manuscript are too blurry to observe the details. Please update them in high-quality vector format.

- The size of some tables (Tab.2 and Tab.3) seems to exceed the page margin.

- The grammar of this manuscript needs to be further polished.

Comments on the Quality of English Language

Need to be further polished.

Reviewer 3 Report

Comments and Suggestions for Authors

The manuscript “Research on Image Stitching Algorithm Based on Point-Line Consistency and Local Edge Feature Constraints” is fair good one but I would recommend the authors to revisit/rethink several points:

  1.  Even if in Section I, lines 88-104 the authors explained the manuscripts main aspects, I feel that they should explain better what novelty does the proposed solution bring.
  2. Please add a paragraph in Section I that outlines the manuscript itself.
  3. In Section 3 the authors describe each step of the algorithm that is described in Fig. 1, but I would recommend to add an outline of the proposed algorithm and parameters it uses. It would help the readers understand of each step. For me it was hard to follow what are the order of the steps.
  4. Please redo Figure 10, so any readers can understand the axis.
  5. I believe that the manuscript would benefit from more examples of stitching done by the proposed solution.
  6. Please revise the entire manuscript for several small mistakes like in title 3.2.2.

Round 2

Reviewer 1 Report

Comments and Suggestions for Authors

I thank the authors for addressing my concerns marked as Comments 1, 2, 5, 6, 7, 8, and 9. However I do not feel Comment 3 on the clarity of Sections 3.2 and 3.3, or Comment 4 on the use of Multi-GS in Section 3.4, have been addressed at all. Sections 3.2 and 3.3 have only been superficially rephrased by replacing some words with synonyms and shuffling words and sentence clauses around. The logical flow and level of clarity have not been improved. I have now tried to “read between the lines” to understand the algorithm in order to provide specific concerns, but I still believe these sections need major structural rework. I apologize for providing such a large list of comments in this second round, but I had hoped the authors would take it upon themselves to improve these sections without explicit notes. On a positive note, the authors’ summary in Response 3 to Reviewer 1 beginning with “description of key steps” is very coherent and precise; I wish the rest of the manuscript was this clear.

Overall:

  1. Figure 1: Text is not in English (it was correct in version 1).

  2. Terminology is not consistent. For example the following terms all appear to refer to the same thing: contour, edge, curve, spindle (but this is not the only example). Pick one word or phrase for each concept and use it throughout.

Section 3.2:

  1. 249-256: Don’t need to explain how to establish a “minimum bounding box” as this is a well-established concept.

  2. 263: “the preset distance threshold” - I think “a” was meant rather than “the”, to indicate this parameter is being introduced here rather than referring to something established earlier. It may seem like a minor grammatical quibble but the difference is important. This sort of issue is common throughout the text making it difficult to read accurately.

  3. 277: What are the requirements for “the required regular shape contours”? These are not stated.

  4. 284-286: Peak detection has a deep prior literature, and this is perhaps the most naive possible implementation. Figure 5 even shows what should probably be considered spurious peaks due to very small changes in edge direction. A more sophisticated peak detection algorithm would fix this.

  5. 287: What’s a “spindle” here? This word is never used again in the manuscript.

  6. 293: It looks like a sentence was cut off around “the local edge Similar structures exist” (after “edge”).

  7. 311: Is this threshold parameter the same one mentioned on line 296, or a new one?

  8. 318-319: Don’t need to explain what a polynomial is.

  9. 319: What polynomial order n was selected, and why?

  10. 324-325: Don’t need to define Residual Sum of Squares.

  11. Figure 6(c): I believe Section 3.2.2 should reference panel C as an example of the results of the contour merging operation.

Section 3.3:

  1. 336: As a homography only maps a plane to a plane, how robust is this step in the face of non-coplanar structures in the image? Addressing this issue was one of the strengths of Multi-GS, so why is that algorithm not being leveraged here?

  2. 338: Which “matching feature pairs” are used to calculate the homography?

  3. 347: What are mask_refer and mask_target? They were never defined.

  4. 349: Don’t need Equation 3-8 for this, the text description is enough.

  5. 356-373: Is edge contour filtering also applied to the target image? If not, why?

  6. 363: L(e_i) is defined but never referenced again. Why was it defined?

  7. 364: should reference formula 3-4, not 3-7.

  8. 366: Another cut off sentence “considered to be The two edge” (after “be”).

  9. 370: In the phrase “delete the relationship between the remaining edge objects in the edge object set and the edge object”, is this referring to a row in a distance matrix? It would be clearer to describe this whole section as operating on a distance matrix.

  10. 403-404: What are “partially matched edge contours?” The purpose of this paragraph is unclear without that definition.

  11. 413: What does “their” refer to? Edge contours?

  12. 414-419: This paragraph does not provide enough details for this step. What are the “structure type” classes used? There is another cut off sentence on line 417. What is done with the results of the “same structure type” determination and “similarity of the two edges” in order to affect the contour edge matching between image pairs?

  13. It would be great to see more image examples with curved contours. The top panel of Figure 7 does have a few curves, but perhaps the other panels could be replaced with examples that show curved contours rather than all straight lines. Otherwise the improvement over Jia et al. is not obvious.

Section 3.4:

  1. Section 3.4 has been cut down dramatically but there is still no mention of Multi-GS. The previous version of the text appeared to be a rough explanation of Multi-GS, but most of that text has been removed in this version. Is this pre-alignment step directly using the Multi-GS method, or something different but related that the authors derived themselves? If it’s just Multi-GS then say so, along with a clearer explanation of how the fitting problem is posed in terms of the point-line and edge contour features. If, however, this is a new method then it requires a much more thorough explanation. Furthermore, the authors should explain why they are not just using Multi-GS which seems well-suited to this exact problem.

Section 4.1.1:

  1. Having read Sections 3.2 and 3.3 in more detail, I have found five different threshold parameters mentioned: contour distance (263), spindle length (287), edge density (296), polynomial fit (328), and edge close distance threshold T (364). However line 529 only mentions two threshold parameters when describing the experimental setup. What about the others?

  2. The data shown in Tables 2 and 3 and Figure 9 do not support the claim that the contour constraints provide a higher quality image alignment. The constraints=Yes values are not even consistently lower (better) than the constraints=No values. 

Section 4.2.4:

  1. Why are the SSIM results only reported on a subset of the datasets that RMSE was reported on?

  2. Figures 12 & 13: Yellow and pink lines and points are misaligned. It looks like the yellow points are on the pink line, and the yellow line has no points. Yellow-on-white is very hard to see and a darker color should be used, anyway.

Comments on the Quality of English Language

The English was understandable overall, but Sections 3.2-3.4 need more attention. The text has frequent passages written in the imperative ("Merge the data...") whereas most of the text uses the preferred passive voice ("The data were merged..."). All instances of imperative voice should be corrected.

Reviewer 2 Report

Comments and Suggestions for Authors

Thanks for the authors' detailed rebuttal. Most of my concerns have been well addressed.

Reviewer 3 Report

Comments and Suggestions for Authors

Dear authors, 

First of all I would like to thank you for the revision done in the manuscript! However, I would ask you to consider the following points:

1. Figure 1 is improved, but I would recommend adding actual flow boxes showing each step, like feature extractor, edge-detection, LSD detection, matching. Visually the blocks make sense but do not show the extent of the proposed algorithm. 

2. Please consider the following manuscript for edge-detection part:

[1] Orhei, Ciprian, et al. "Dilated filters for edge-detection algorithms." Applied Sciences 11.22 (2021): 10716.

Round 3

Reviewer 1 Report

Comments and Suggestions for Authors

I thank the authors for thoroughly addressing all of my comments. I have noted several minor errors in the new text which can be fixed easily without any further rounds of review.

  1. 217 and Fig. 4 legend: “Closed operation” should be “closing operation”.

  2. Multiple locations: “binomial” should be “quadratic”. These are not the same thing. “Quadratic” is the correct term for a second-order polynomial.

  3. 434-438: The two sentences “Firstly, the shape similarity…” and “The method of…” are duplicated two times. Remove the duplicates.

Comments on the Quality of English Language

No further comments.

Reviewer 3 Report

Comments and Suggestions for Authors

Thank you for considering the remarks highlighted in the review. I am overall satisfied with the modifications made by the authors.
